# Analysis of miR-143, miR-1, miR-210 and let-7e Expression in Colorectal Cancer in Relation to Histopathological Features

**DOI:** 10.3390/genes13050875

**Published:** 2022-05-13

**Authors:** Hanna Romanowicz, Piotr Hogendorf, Alicja Majos, Adam Durczyński, Dariusz Wojtasik, Beata Smolarz

**Affiliations:** 1Laboratory of Cancer Genetics, Department of Pathology, Polish Mother’s Memorial Hospital Research Institute, Rzgowska 281/289, 93-338 Lodz, Poland; hanna-romanowicz@wp.pl; 2Department of General and Transplant Surgery, N. Barlicki Memorial Clinical Hospital, Medical University of Lodz, 90-153 Lodz, Poland; piotr.hogendorf@umed.lodz.pl (P.H.); strzalka.alicja@gmail.com (A.M.); adam.durczynski@umed.lodz.pl (A.D.); 3Hospital in Glowno, 95-015 Glowno, Poland; dwojtasik@vp.pl

**Keywords:** miRNA, colorectal cancer, RT-PCR

## Abstract

Background: MicroRNAs (miRNAs) are small RNA molecules involved in the control of the expression of many genes and are responsible for, among other things, cell death, differentiation and the control of their division. Changes in miRNA expression profiles have been observed in colorectal cancer. This discovery significantly enriches our knowledge of the pathogenesis of colorectal cancer and offers new goals in diagnostics and therapy. Aim: The aim of this study was to analyze the expression of four miRNA sequences—miR-143, miR-1, miR-210 and let-7e—and to investigate their significance in the risk of developing colorectal cancer. Materials and methods: miRNA sequences were investigated in formalin-fixed, paraffin-embedded (FFPE) tissue in colorectal cancer patients (*n* = 150) and in cancer-free controls (*n* = 150). The real-time PCR method was used. Results: This study revealed a lower expression of miR-143 in colorectal cancer patients than in the controls. miR-143 was positively correlated with the degree of tumor differentiation (grading). Three out of four analyzed miRNA (miR-1, miR-210 and let-7e) were found to be statistically insignificant in terms of colorectal carcinoma risk. Conclusions: miR-143 may be associated with the development of colorectal cancer.

## 1. Introduction

The infamously high position of colorectal carcinoma (CRC) among the most common and deadly cancers worldwide seems to be stable. Due to its commonness and relatively early-appearing symptoms, which have enabled research and drug development, patients can benefit from numerous lines of treatment availability. Sadly, these developments have not been equally advantageous for all patients. About one quarter of cases are still diagnosed in stage IV with unsatisfactory 5-year overall survival (OS) lower than 30% [1].

The early detection of the disease seems to be the key to improve treatment outcomes. Although validated screening methods such as colonoscopy, sigmoidoscopy, CT colonography, multitarget stool DNA tests, and fecal immunochemical tests have been proven to reduce CRC incidence and mortality [2], their use does not solve the problem of diagnostic delay in patients presenting symptoms. A more accurate definition of metastasizing potential, treatment susceptibility, and OS within the different stages at the time of diagnosis could additionally help to extend OS and improve the quality of life in CRC patients.

Therefore, various MicroRNAs (miRNAs—small, noncoding RNAs) constitute the object of research in all branches of oncology. One MiRNA can influence the translation of many proteins through binding to target mRNAs, thus causing multiple effects. Some of them are potentially promising medication grip points [3,4,5]. 

Due to their small size, miRNAs are characterized by high stability in different kinds of samples, e.g., serum, frozen tissues and, especially important because of the availability of this material, formalin-fixed paraffin-embedded tissues [6]. 

Until now, at least 69 miRNAs dysregulated in CRC have been detected [7]. 

Recently, the determination of more promising miRNAs has been intensively studied in relation to colorectal cancer. Survival-dependent varied miRNAs (sDMIR) have been detected. In their publication, Wang et al. identified three DMIR values (hsa-miR-21-3p, hsa-miR-194-3p and hsa-miR-891a-5p) correlated with the most important prognostic values of CRC patients. High expression levels of hsa-miR-21-3p and hsa-miR-194-3p have been shown to be associated with early T stages. However, hsa-miR-891a-5p showed the opposite result [8].

Recent data indicate that miRNAs can be used as biomarkers for the early diagnosis and prediction of CRCs. They may also have potential therapeutic uses [9].

The objects of interest of this study were the following miRNAs: miR-143/miR-145, miR-1 and miR-210, let-7e. 

The miR-143/miR-145 cluster is known to act as a tumor suppressor, with targets recognized in both CRC (miR-143: *MACC1*, *TLR2*; miR-145: *Cateninδ-1*, *DFF45*, *STAT1*; miR-143 and miR-145: *KLF5*) and other malignancies (miR-143: *KRAS*—CRC, prostate cancer, pancreatic cancer; Akt—CRC, bladder cancer, HCC, glioma; Bcl2—CRC, breast cancer, ovarian cancer, bladder cancer; miR-145: VEGF—CRC, breast cancer, ovarian cancer, thyroid cancer, glioblastoma; miR-143 and miR-145: IGFIR) [10,11,12,13,14,15,16,17,18,19,20,21,22,23,24,25,26,27,28,29,30,31]. –6700. It plays its role in the multidrug resistance phenomenon. The ectopic expression of miR-145 (in vitro experiments) increased the sensitivity of CRC cells to vemurafenib [Peng] 5-FU, irinotecan and oxaliplatin [32,33]; it was also the object of studies on drug design—both in vitro and in vivo—with promising effects [15,33,34,35,36,37,38].

miR-1 inhibits tumor growth and metastasis by simultaneously targeting multiple genes: CDK4 (cyclin-dependent kinase 4), TWF1 (twinfilin actin-binding protein 1), WASF2 (WAS protein family, member 2), CNN3 (calponin 3, acidic), CORO1C (coronin, actin binding protein, 1C) and TMSB4X (thymosin beta 4, X-linked) [39]. 

Its downregulation has been reported in many cancers (CRC, gastric cancer, medulloblastoma, breast cancer, prostate cancer, etc.), and its expression has been proved to be negatively correlated with VEGF expression in CRC tissues [40,41]. 

The overexpression of miR-210 through the upregulation of pro-apoptotic Bim expression and Caspase 2 induces reactive oxygen species generation and apoptosis in CRC cells [42]. 

It also regulates the JAK-STAT signal transduction pathway by targeting PIAS4, influencing breast cancer chemosensitivity, and has shown positive correlation with the chemoresistance of breast cancer MCF-7 cells [43].

let-7e is a tumor suppressor that is down-regulated—partially by the elevated expression of IGF1R—in CRC cells. Raising its concentration reduces IGF1R expression as well as the cell proliferation, migration and invasion of colorectal cancer cells [44]. It is also described to be involved in cell differentiation, as well as to influence the MAPK, mTOR, FoxO and p53 signaling pathways [45].

The aim of this study was to analyze the level of expression of all four above-mentioned miRNA in colorectal cancer and to correlate the findings with pathological data and to align them with the risk of colorectal cancer.

## 2. Material and Methods

### 2.1. Patients

The research material consisted of tumor specimens collected from patients with colorectal cancer (*n* = 150) treated surgically at the Hospital in Glowno in the years 2015–2021. The material for the study came from paraffin blocks obtained from postoperative material. 

Histopathological examinations were carried out in the Department of Pathology of the Polish Mother’s Memorial Hospital Research Institute (Table 1). The degree of differentiation of the tumor was determined according to the scale from G1–G2 (G-grade). The clinical stage of colon cancer according to the Dukes’ classification was determined.

DNA from normal colorectal tissue (*n* = 150) served as the control. Table 2 shows the number and age of patients and the control group. Genetic studies of miRNA expression levels were carried out in the Laboratory of Cancer Genetics Polish Mother’s Memorial Hospital Research Institute in Lodz. The Bioethical Committee of the Lodz Medical University approved the study (approval number RNN/266/21/KE).

### 2.2. RT-PCR Methods

The first stage of the study was to evaluate the expression of micro-RNA molecules using RT-PCR in cancer tissues derived from archival paraffin blocks (FFPE, formalin-fixed, paraffin-embedded tissue) from patients. From each paraffin block, 2 scraps with an area of approx. 1 × 1 cm and a thickness of 5–8 μm were cut and placed in an Eppendorf tube. 

The total RNA was extracted from the paraffin-fixed tissues (FFPEs) using the High Pure miRNA isolation kit (Roche Diagnostics GmbH, Mannheim, Germany), in accordance with the manufacturer’s instructions. The FFPE samples were placed in 2 mL of Eppendorf tubes, dewaxed with 100% xylene, washed with 100% ethanol and dried at 55 °C for 10 min. Dried tissue was suspended in 100 μL of buffer for the lysis of paraffin tissue (included in the set) and digested with proteinase K at 55 °C overnight. The resulting total RNA was immediately used for cDNA synthesis, or stored at −80 °C prior to use.

Reverse transcription was performed using the Maxima First Strand cDNA synthesis kit (Thermo Fisher Scientific, Inc., Waltham, MA, USA) according to the manufacturer’s protocols. An amount of 500 ng of total RNA was used as the starting material, and the reverse transcription was carried out under conditions optimized for use with this set (25 °C for 10 min, 50 °C for 30 min, 85 °C for 5 min). cDNA samples were stored at −20 °C. The quantification of the miRNA was performed using TaqMan™. In addition, the GADPH assay was used as an endogenous control. qPCR reactions were carried out in a reaction volume of 10 μL, including 10 ng cDNA, 5 μL TaqMan Fast Advanced PCR Master Mix and 0.5 μL of a suitable starter (20×). Samples were incubated in a 96-well plate at 95 °C for 3 min, followed by 40 cycles of 95 °C for 1 s and 60 °C for 20 s. Relative level of expression was determined by the 2^−ΔΔCq^ method. MiRNA expression was correlated with clinical–histopathological factors.

### 2.3. Statistical Analysis

Applied Biosystems Data Assist Software V3.01 Thermo Fisher Scientific—US (Warsaw, Poland) was used to statistically process the results (https://www.thermofisher.com/pl/en/home/technical-esources/softwaredownloads/dataassist-software.html) (accessed on 22 May 2021).

The significance of the differences was analyzed at the level of gene expression and mRNA using non-parametric tests (Mann–Whitney U test and Kruskal–Wallis test) for the lack of normality of the distribution of the obtained results, which was confirmed by the Shapiro–Wilks test. The R-Spearman test was used to assess the correlation between the variables. Statistical significance was confirmed at *p* < 0.05. Receiver operating characteristic curves (ROCs) were created and the area under the curve (AUC) was calculated to assess the specificity and sensitivity of the selected miRNAs.

## 3. Results

The expression results of miR-143, miR-1, miR-210, let-7e in colorectal cancer patients and controls are presented in Table 3 and Table 4, respectively.

Our study found no statistically significant differences in the expression of miR-1, miR-210 or let-7e between colorectal cancer patients and the control group. However, we found a statistically significant lower expression of miR-143 in cancer cases compared to the controls (Figure 1). Table 5 shows comparisons of the expression of the study miRNA in the patients and the controls.

The expression of the miRNA sequences miR-143/miR-145, miR-1, miR-210 and let-7e was also analyzed statistically for correlation with the clinical and histopathological data regarding age, sex, grading, staging and TNM. The study found a statistically significant correlation between the degree of tumor differentiation (grading) and miRNA-143 in the patient group: it was positively correlated with the degree of tumor differentiation (Table 6 and Table 7 and Figure 2). All other above-mentioned correlations proved to be statistically insignificant. ROC curves were constructed to compare colorectal cancer and the positive lymph nodes of colorectal cancer. The results indicated that the absence of miR-143 is characteristic of metastatic tissue derived from the positive lymph nodes of colorectal cancer (Figure 3).

## 4. Discussion

Cancer is an important civilization problem of our age. Work is still underway to search for new factors responsible for the carcinogenesis process. In the introduction, we cited various current literature data to provide compelling evidence that miRNA transcription plays an important role in key biological processes in colorectal cancer. 

There are reports in the scientific press suggesting a link between micro-RNA expression and prognosis, survival in patients, and the advancement of the cancer process [46,47,48,49]. In this paper, an attempt was made to verify histopathological data in relation to micro-RNA in colorectal cancer. In the context of miRNA expression levels, the following were assessed: the clinical stage of colorectal cancer, the presence of metastases in the lymph nodes, and the degree of histological malignancy of the tumor. An interesting aspect of such an assessment is whether microRNA can serve as a negative prognostic factor in a group of patients affected by colorectal cancer. On this basis, it is possible to identify patients belonging to the risk group whose prognosis would be worse and the therapeutic process should be more aggressive, even in the case of cancer at an early clinical stage.

In colorectal cancer cells, the increased expression of selected miRNAs is more often observed. This fact indicates that they are more often oncogenic in nature [50]. Increased miRNA expression may result from the amplification of genes encoding miRNAs, from more efficient biogenesis, constitutive activity of their promoters, or the greater stability of miRNA molecules [51].

Increased miRNA expression leads to the repression of numerous genes with suppressor activity.

In our study, we presented an analysis of the expression of miR-143/miR-145, miR-1, miR-210 and let-7e in patients with colorectal cancer compared to non-cancer controls, with the aim of possibly correlating these miRNAs with the risk of the above-mentioned malignant tumor. The only miRNA with which our statistical analysis showed a significant correlation with colorectal cancer was miR-143. In some cases, this sequence showed a clear downward adjustment. miR-143 was found to be significant in terms of the interconnections with the pathological data of cases. It also correlated with the degree of differentiation of the tumor. 

We selected miRNAs for research based on the literature data. According to the available literature, miR-143 and miR-145 (miR-143/145), which are in the same cluster within the chromosomal region 5q32–33, have negative correlations with many types of cancer, including CRC [52,53,54,55]. An in vitro model of colorectal cancer showed a decrease in miR-143 expression, whose target gene is, among others, the Raf1 oncogene [56]. The decreased expression of miR-143 in colorectal cancer cells results in the increased activity of methyltransferase 3A DNA (DNMT3A) and the increased proliferation of cancer cells [57].

MiRNA mimics the RNA-induced silencing complex (RISC) and inhibits the downstream target mRNAs. A growing body of evidence points to the effectiveness of miRNA therapy in in vitro and in vivo models. For example, the miR-143/145 cluster is a tumor suppressor that is often down-regulated in several tumors, including CRCs. MiR-143/145 directly targets K-RAS and the insulin receptor substrate. Therefore, the ectopic expression of the miR-143/145 cluster after the transfection of its mimics reduces the migration and invasiveness of the CRC cells [58].

The overexpression of miR-143 inhibits tumor growth and increases chemosensitivity to oxaliplatin. Mutations in the PI3K/AKT pathway are also associated with resistance to oxaliplatin. Before the PI3K/AKT pathway, IGF-1R is inhibited by miR-143, inactivating the AKT signaling pathway and lowering HIF-1α levels [58].

The next sequences we studied were miR-1, miR-210 and let-7e. To reduce the expression of let-7, miR-34, miR-342, miR345, miR-9, miR-129 and miR-137, and thus the development of colorectal cancer, leads to the hypermethylation of the promoters of these miRNAs [59].

It is worth mentioning that miR-1 targets VEGFA, which is the main mediator of angiogenesis. A potential target in 3′ UTR VEGFA that can interact with miR-1 has been detected by Zhu et al. [41]. MiR-1 has been shown to suppress VEGFA expression and may affect the MAPK and PI3K/AKT pathways. miR-210 is included in autophagy in cancer. There has been an increase in HIF-1α and miR-210 under hypoxia conditions, and miR-210 may cause accelerated cellular autophagy [60]. IGF1R is a direct target for let-7e members. Let-7, a representative miRNA that acts as a tumor suppressor, is poorly expressed in malignant tumors [58]. Let-7e has been reported to increase the radiation sensitivity of CRC cells by directly suppressing IGF1R [61].

In our studies, miR-1, miR-210 and let-7e were found to be statistically insignificant from the point of view of colorectal cancer risk. Our results are in contrast to the literature data that support the role of these miRNAs in tumorigenesis. This may be due to the fact that there are some obvious limitations to our study that need to be mentioned and clarified. The dominant drawback of our analysis is the numbers: the introduction of 150 cases and 150 controls (a total of 300 tests) makes an experienced researcher cautiously and skeptically draw final conclusions. For genetic studies, the analyzed patient groups may simply be quantitatively unsatisfactory to make a fair assessment. In addition, little is known still about the role of “transcription noise” and how to properly design a study to draw definitive and defensible conclusions. The results of the study lead to the conclusion that studies of selected miRNA sequences are supported by convincing premises, and research in this area must be continued.

Recent studies have shown that good candidates for predicting and diagnosing gastrointestinal (stomach and colon) cancers may be single-nucleotide polymorphisms (SNPs) and miRNA profiles [62].

Significant polymorphisms in the diagnosis and prognosis of gastrointestinal cancers include the rs141178472 polymorphism of PIK3CA, rs12373 PAUF, rs13347 CD44, rs6504593 and rs1049109 in 3’ UTR IGF2BP1, rs1590 TGFBR1, rs4939827 MLH329861, rs12915554 GREM1, rs74693964 KRAS, rs10889677 IL-23R, rs3748067 IL-17A, rs12537 MTMR3, rs3202538 ERBB3, rs56288038 IRF-1 and rs9005 IL-1RN. The differentiated expressions of key miRNAs have been shown to target SNPs sites. 

The following miRNAs have been identified as strong post-transcriptional regulators: miR-520a: PIK3CA; miR-571: PAUF; miR-509-3p: CD44; miR-21: IGF2BP1; miR532-5p: TGFBR1; miR-193a-3p: MLH3; miR-375: SMAD7; miR-185 -3p: GREM1; miR-145 and miR-143: KRAS; miR-10a-3p: IL-17A; miR-181a: MTMR3; miR-204 and miR-211: ERBB3; miR-502-5p: IRF-1; miR-197: IL-1F5; and miR-148a: SCRN1. PAUF rs12373: miR-571, MLH3 rs108621: miR-193a-3p and SMAD7 rs4939827: miR-375 have been confirmed as biomarkers in the prediction, early prognosis, diagnosis and monitoring of gastrointestinal cancers [62].

Micro-RNA molecules are considered to be extremely stable; it is possible to evaluate them in frozen tissues, serum and, above all, in histopathological material derived from paraffin blocks [63,64,65,66,67]. In our research, we confirmed the usefulness of such material in micro-RNA research.

With all of the above findings in mind, and recognizing the limitations of our study, we dare say that this research has shed new light on miRNAs in colorectal cancer and contributes to a growing—but still unclear—knowledge of these non-coding sequences in oncology.

## 5. Conclusions

MiRNA-143 may be a potential prognostic factor in colorectal cancer. However, further research is necessary to unequivocally confirm such a hypothesis.

## Figures and Tables

**Figure 1 genes-13-00875-f001:**
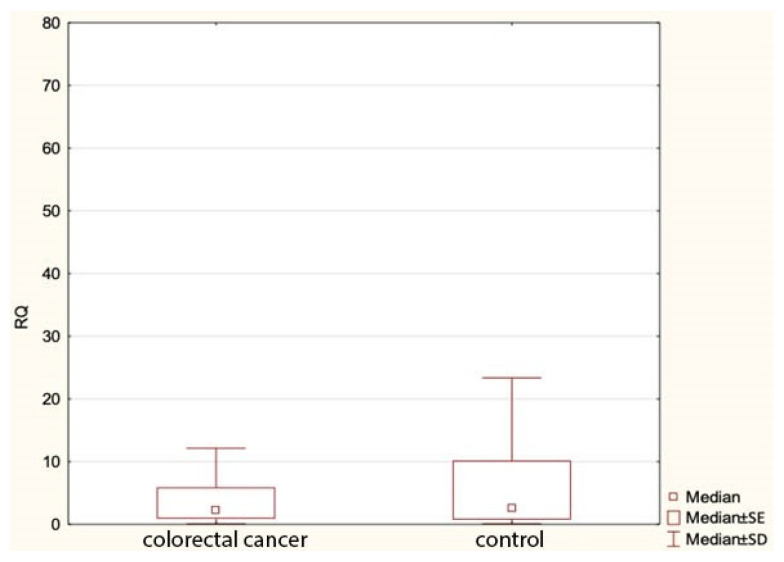
Expression of miR143 in colorectal cancer group and controls. RQ—relative quantification.

**Figure 2 genes-13-00875-f002:**
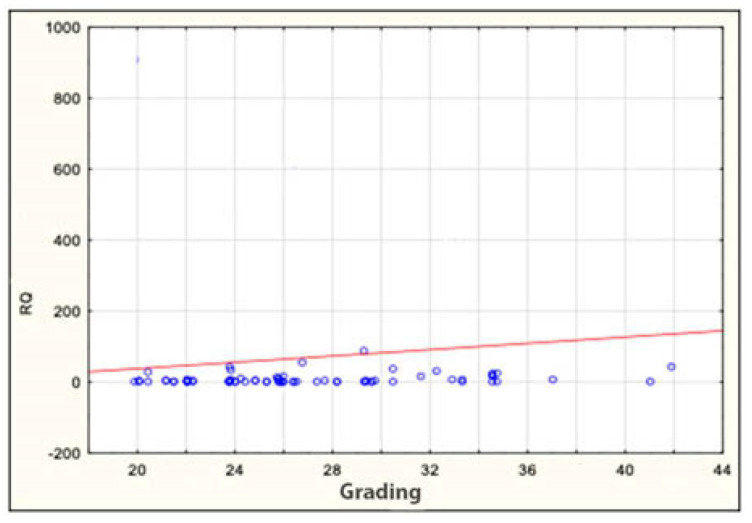
Correlation between miRNA-143 and grading in patients. RQ—relative quantification.

**Figure 3 genes-13-00875-f003:**
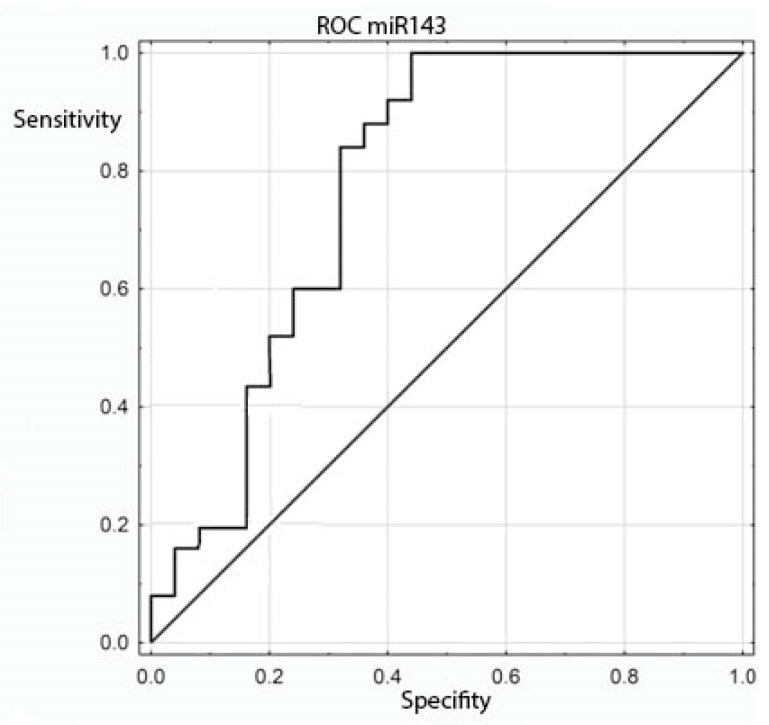
ROC curve for the ability of miR-143 to differentiate between colorectal cancer and positive colorectal cancer lymph nodes.

**Table 1 genes-13-00875-t001:** Histopathological characteristics of colorectal cancer patients.

	Number of Cases (%)
Dukes’ stages	
Stage A	51 (34)
Stage B	69 (46)
Stage C	18 (12)
Stage D	12 (8)
Tumor size	
T1	78 (52)
T2	60 (40)
T3	12 (8)
Lymph node status	
N0	78 (52)
N1	50 (33)
N2	12 (24)
N3	10 (7)
Grading	
G1	68 (45)
G2	43 (29)
G3	39 (26)

**Table 2 genes-13-00875-t002:** The number and age of the patients and the control group who underwent miRNA tests.

Patients (*n* = 150)		Controls (*n* = 150)	
Women (*n* = 65)	Men (*n* = 85)	Women (*n* = 68)	Men (*n* = 82)
Age (range)	Age (range)	Age (range)	Age (range)
56–68 years	53–70 years	54–67 years	54–67 years
Age (average)	Age (average)	Age (average)	Age (average)
62.1 ± 10.2	61.3 ± 11.2	60.1 ± 10.3	62.3 ± 10.2

**Table 3 genes-13-00875-t003:** Expression of miRNA in patients.

miRNA	N	RQ Mean	RQ Min.	RQ Max.	SD
let-7e	150	6.65	0.063	28.33	21.63
miR-1	150	2.96	0.002	92.28	7.40
miR-210	150	2.81	0.072	41.78	2.84
miR-143	150	2.35	0.065	43.53	4.67

**Table 4 genes-13-00875-t004:** Expression of miRNA in controls.

miRNA	N	RQ Mean	RQ Min.	RQ Max.	SD
let-7e	150	6.65	0.063	28.33	21.63
miR-1	150	2.96	0.002	92.28	7.40
miR-210	150	2.81	0.072	41.78	2.84
miR-143	150	2.35	0.065	43.53	4.67

**Table 5 genes-13-00875-t005:** Correlation of expressions of the studied miRNA in patients and controls.

	Mann–Whitney *U* Test Relative to the Variable Zmn5 Bold Results Are Important *p* < 0.05000								
miRNA	Rank Sum—Cases	Rank Sum—Controls	*U*	Z	*p*	Z	*p*	N—Cases	N—Controls
let-7e	12350.00	7540.000	3200.000	−1.38	0.125	−1.39	0.124	150	150
miR-1	11232.40	7865.500	4282.500	−1.67	0.064	−1.87	0.063	150	150
miR-210	11652.00	6328.000	3078.000	1.65	0.070	1.72	0.082	150	150
miR-143	11284.00	9806.00	2824.000	**−2.14**	**0.030**	**−2.14**	**0.030**	150	150

**Table 6 genes-13-00875-t006:** Correlation between studied miRNA and the degree of tumor differentiation (grading).

Variables	Spearman’s Correlation *p* < 0.05000			
	N—Patients	R—Spearman	The Two-Sample *t*-Test T (N − 2)	*p*
let-7e and grading	150	0.141	1.20	0.063
miR-1 and grading	150	0.102	1.71	0.078
miR-210 and grading	150	0.110	0.86	0.224
miR-143 and grading	150	**0.250**	**2.265**	**0.023**

**Table 7 genes-13-00875-t007:** Correlation between studied miRNA and the stage of clinical advancement of colorectal cancer (Dukes’ classification).

Variables	Spearman’s Correlation *p* < 0.05000			
	N Patients	R—Spearman	The Two-Sample *t*-Test T (N − 2)	*p*
let-7e and Dukes’ stages	150	−0.074701	−0.074	0.356
miR-1 and Dukes’ stages	150	−0.108	−0.110	0.220
miR-210 and Dukes’ stages	150	−0.011	−0.011	0.788
miR-143 and Dukes’ stages	150	−0.011	−0.012	0.705

## Data Availability

All data and materials, as well as software application, support the published claims and comply with field standards.

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
