# Peer review of "Analysis of miR-143, miR-1, miR-210 and let-7e Expression in Colorectal Cancer in Relation to Histopathological Features"

_genes, 2022, doi:10.3390/genes13050875_

Round 1

Reviewer 1 Report

  1. In abstract: explain the other microRNA in the result section.
  2. In the discussion and introduction use new references like:

PMC8386531, PMID: 34550619, PMID: 33567635, PMID: 35330456

  1. Table3: what is RQ, explain it as the caption.
  2. The quality of the figure is low.
  3. Why did the author select these microRNA, what is your rationale?
  4. Analysis of the microRNA based on the ROC analysis and draw roc curve.
  5. What is your novelty?

Author Response

Thank you for your review.

I would like to kindly ask you to reconsider the publication of our revised paper:

" Analysis of miR-143, miR-1, miR-210 and let-7e expression in colorectal cancer in relation to histopathological features”.

I hereby provide responses to the reviewers and list the changes that have been made in the revised version of our paper.

  1. In abstract: explain the other microRNA in the result section.

A sentence of explanation has been included in the abstract

  1. In the discussion and introduction use new references like:

PMC8386531, PMID: 34550619, PMID: 33567635, PMID: 35330456

  Articles have been included in the text 

  1. Table3: what is RQ, explain it as the caption.

It is explained in the description of Figure

  1. The quality of the figure is low.

Figure quality has been improved

  1. Why did the author select these microRNA, what is your rationale?

We tried to give an explanation in the introduction of the work. We selected miRNAs based on literature data and their importance in colorectal cancer. In addition, these miRNAs have not been studied in the Polish population in colorectal cancer.

6.Analysis of the microRNA based on the ROC analysis and draw roc curve.

We have added additional analysis on Figure 2.

  1. What is your novelty?

Although, as outlined in the article, numerous miRNA sequences are somehow correlated with colorectal cancer, the exact meaning and context of this interaction remains unclear and requires clarification. This work aims to expand the still unclear and incomplete knowledge about the role of selected miRNA sequences in CRC. Moreover, these studies are innovative in the Polish population. They have not yet been conducted in this aspect.

I hope you find our revised Manuscript satisfying so that it can meet the criteria of publication in your Journal.

Looking forward to hearing from you,

Yours sincerely,

Beata Smolarz

Reviewer 2 Report

The manuscript is interesting.

In line 185, please, explain the relation of this investigation and endometrial cancer.

Conclusions should be more specific, please.

Author Response

Thank you for your review.

I would like to kindly ask you to reconsider the publication of our revised paper:

" Analysis of miR-143, miR-1, miR-210 and let-7e expression in colorectal cancer in relation to histopathological features”.

I hereby provide responses to the reviewers and list the changes that have been made in the revised version of our paper.

The manuscript is interesting.

In line 185, please, explain the relation of this investigation and endometrial cancer.

Sorry, an error occurred. There should be colon cancer

Conclusions should be more specific, please.

Conclusion has been corrected

I hope you find our revised Manuscript satisfying so that it can meet the criteria of publication in your Journal.

Looking forward to hearing from you,

Yours sincerely,

Beata Smolarz

Round 2

Reviewer 1 Report

1. Roc curve analysis did not perform by the author and they just analyzed based on the correlation.

Author Response

Thank you for your review.

Roc curve analysis have been added (Figure 3)

I hope you find our revised Manuscript satisfying so that it can meet the criteria of publication in your Journal.

Looking forward to hearing from you,

Yours sincerely,

Beata Smolarz